# Why Was the Perception of Human Rights Respect and Care Satisfaction So High in Users of Italian Mental Health Services during the COVID-19 Pandemic?

**DOI:** 10.3390/jcm11040997

**Published:** 2022-02-14

**Authors:** Mauro G. Carta, Federica Sancassiani, Ruben Ganassi, Paola Melis, Silvia D’Oca, Michela Atzeni, Fermanda Velluzzi, Caterina Ferreli, Laura Atzori, Cesar I. Aviles-Gonzales, Maria Roberta Serrentino, Matthias C. Angermeyer, Giulia Cossu

**Affiliations:** 1Department of Medical Sciences and Public Health, University of Cagliari, SS 554, 09042 Monserrato, CA, Italy; federica.sancassiani@unica.it (F.S.); rubenganassi1@gmail.com (R.G.); pirimpola@hotmail.com (P.M.); silvia.doca13@gmail.com (S.D.); michela.atzeni93@gmail.com (M.A.); fernandavelluzzi@gmail.com (F.V.); ferreli@unica.it (C.F.); atzoril@unica.it (L.A.); infermiere2010@gmail.com (C.I.A.-G.); roberta.serrentino@gmail.com (M.R.S.); giuliaci@hotmail.com (G.C.); 2Center for Public Mental Health, Untere Zeile 13, A-3482 Gösing am Wagram, Austria; angermeyer@aon.at

**Keywords:** user satisfaction, mental health, quality of care, human rights, COVID-19 pandemic

## Abstract

The aim of this study was to compare users’ and mental health workers’ (MHW) perception of respect of human rights and job/care satisfaction in mental health services in Italy during the COVID-19 pandemic. A sample of users and MHW of Sardinia, Italy, fulfilled the “Well-Being at work and respect for human rights questionnaire” (WWRR). The study included 240 MHW and 200 users. Users showed a higher level of satisfaction of care than MHW of work, and a higher perception of the satisfaction of users and human rights respected for health workers. Both user and MHW responses were about 85% of the maximum score, except for satisfaction with resources. Responses were higher for users, but users and MHW both showed high levels of satisfaction. In previous surveys, MHW of Sardinia showed higher scores in all items of WWRR, except for satisfaction with resources, compared with workers from other health sectors of the same region, and with MHW from other countries. The low score for satisfaction with resources (in users and staff) is consistent with a progressive impoverishment of resources for mental health care in Italy. The study, although confirming the validity of the Italian model, fully oriented towards community, sets off an alarm bell on the risks resulting from the decrease in resources.

## 1. Introduction

The right of people with psychosocial disabilities to fully participate in the political and public life is constantly challenged by stigma and discrimination; therefore, human rights violations committed against people with mental health issues have been declared a “global emergency” and an “unresolved global crisis” [1,2]. How care is delivered in mental health and how people in need of care perceive the quality of care provided and the respect of their rights in health care facilities is a key issue in addressing stigma and discrimination in mental health [3,4]. In fact, the relationship between the quality of care in mental health care services and the respect of the human rights of users is related bidirectionally: human rights violations negatively impact the quality of care in terms of mental health but, conversely, a high level of respect of human rights can improve the quality of mental health care [1,5].

The climate of respect for the people in care systems affects not only the well-being of users but also that of the professionals. The concept according to which the perception of the respect of the human rights of users by staff is a main component of organizational well-being in mental health care services is increasingly affirming [6,7,8]. The starting hypothesis was that the organizational well-being of a health care agency is the complex result of mutually influencing components. That is, the non-respect of the rights of a system component (the users) creates discomfort in the users, conflict between users and MHW, and alack of esteem of the professionals by the users; this therefore ends up reverberating also in the well-being of the MHW.

This concept was recently confirmed by the results of a multinational study conducted in the Mediterranean area. The mental health workers of three different countries who fulfilled the “Well-Being at work and respect for human rights questionnaire” (WWRR) showed closely related scores in the six main items concerning the perception of the respect of the rights of users and professionals at work, job satisfaction and work organization in mental health facilities [9]. This explains that the higher the score in the items that measured the perception of the respect of human rights, the higher was the perception of job satisfaction and organizational well-being [9]. A comparison between the scores of the same tool fulfilled by mental health workers of four countries in the same Mediterranean area (Italy, North Macedonia, Gaza and Tunisia) shows that Italian workers gave the highest score in the item concerning the perception that users’ rights were most respected in their care services and, consistently, they were the most satisfied with their work [10]. However, they did not give the highest score in item 6 “Staff’s satisfaction with resources for care”.

Although this result may have been undermined by the fact that Italy was a richer country and with a likely efficient health system [11], a study conducted in an Italian region during the COVID-19 pandemic showed, surprisingly, that health workers engaged in mental health showed higher scores, for the same items of the WWRR questionnaire, than those of health workers engaged in the care of outpatients from other sectors (pain therapy, obesity and endocrinology, ophthalmology, dermatology) [12]. The authors therefore suggested that the result could be the consequence of the strong roots in Italy of mental health services in the community, while the other treatments, albeit for outpatients, were still provided in hospitals, and therefore located in the city and not interpenetrated with other services of support to the citizen. The COVID-19 emergency may have accentuated the influence of these factors [12].

However, the results collected so far have only reflected the staff’s point of view. It is therefore relevant to examine whether the point of view of the users of mental health services is coherent with the one of health workers.

If this were true, it would reinforce the hypothesis that Italy’s organizational style can produce a good quality of care. In fact, Italy is the only country in the world that has mental health services fully rooted in the community and has closed psychiatric hospitals [13].

The objective of this study is to compare the users’ and professionals’ points of view about the perception of the respect of users’ and professionals’ rights in mental health facilities, the perception of the job satisfaction of professionals and the satisfaction with the care of users and organizational well-being in mental health workers from an Italian region.

The study was conducted in a time of high pressure on health services due to the COVID-19 pandemic. Although the study was not originally intended to assess the users’ and MHW’ opinions under COVID-19, the stressful condition for mental health services due to the pandemic made the results even more interesting, even if introducing a further variable.

## 2. Methods

### 2.1. Design

The study was a cross-sectional comparison.

### 2.2. Sample

A sample of volunteer MHW and health workers from 4 community mental health centers in South Sardinia was recruited. The interviews were conducted at the collaborating centers with the consent of the head of the mental health network to whom the authorization of the ethics committee had been presented. In the four care agencies the invitation to participate was addressed to all MHWs, over the course of a month (May 2021). The MHW employees from the national health system working in the four agencies, medical professionals and psychologists in training for specialization but with a responsibility of care in the same health facilities, and employees of external agencies to the national health system but who provided professional activity at the four structures were recruited for the study. Of the 262 MHW thus selected, 11 (4.2%) were not contacted because they were absent throughout the month due to illness.

The recruitment of users was conducted on all users admitted for treatment to the 4 services on the same day of the week for 4 weeks. The exclusion criteria for the users were: being on their first visit to the care facility, not being able to provide consent for lawsuits (decree of a magistrate), but the proband could (if legally designated) receive support from his designated supporting assistant “amministratore di sostegno” if he was admitted (eventuality occurred in only one case), and being in a situation of serious crisis that advised against the proposal of consent to the research and the compilation of the questionnaire. With this methodology, 219 users were contacted and asked to participate.

All potential participants were required to sign an informed consent for participation in the research as required by the authorization of the relevant ethics committee (see below). Any further explanation requested was provided by a research contact present in each of the four care services. It was also specified that the data would be stored in an anonymous database and that agreeing to participation would still leave the participant free to abandon the research and not complete the questionnaire if he wanted to.

### 2.3. Study Tools

Participants, after signing a declaration of informed consent, were subjected to the following tools:(a)A questionnaire collecting data about: age, gender, occupational role, and place of employment. The less frequent professions (for example, social worker or nutritionist) were grouped into “other” to save anonymity, for the same reason the diagnosis of the users was not recorded (the intersection with sex and age would have allowed identification).(b)The tool “Well-Being at work and respect for human rights questionnaire” (WWRR) [9], that was adopted both in the patient and user versions. The WWRR was conceived with reference to the principles of the initiative QualityRights promoted by the World Health Organization for the implementation of the Convention on the Rights of Persons with Disabilities (CRPD) on psychosocial disability, [4,14,15,16]. WWRR measures how users and health workers perceive the respect of human rights (of both patients and staff) in the health services and if this perception is associated with organizational climate at work and the well-being of persons.

The questionnaire was conceived in consultation with expert users, psychiatrists, medical doctors, professionals of rehabilitation, and psychometrists. The original version was developed in Italian and English, but translation in several languages is also available [10]. The five core items are coded according to a Likert scale 1–6, in which score 1 means “Not satisfied at all” and score 6 “Completely satisfied”. Item 6 (“How do you evaluate the current state of care in mental health in your service/ward, with reference to resources?” [9]) is coded in a Likert 1–5 five, score 1 meaning that the resources are completely adequate, and score 5 “Poor assistance is provided due to serious resource deficits” [9]. Item 7 asks about which types of health professionals the interviewee thinks would be most useful to add in your service with only one possible to be admitted. The tool was described in detail in the paper about validation [10].

### 2.4. Statistical Analysis

Statistical analysis was carried out comparing the mean score of responses to the items between health workers and users. The analysis was carried out by one-way ANOVA. The nominal answers to item 7, as well as the comparison between the descriptive statistics of the sample, were analyzed by means of non-parametric statistic such as Chi square (with Yates’s correction if necessary). Odds ratio 95% confidence intervals (OR 95% CI) were also calculated. All analyses were performed using SAS (version 9.4 for Windows).

## 3. Ethics

The Independent Ethical Board of the University Hospital (“Azienda Mista Ospedaliero Universitaria”) of Cagliari, Italy approved the study. The survey was conducted according to the 1995 Declaration of Helsinki and its following revisions [17].

## 4. Results

The study sample included 440 interviewees, of which 240 were mental health workers and 200 users of the same four facilities.

Of the 251 MHWs contacted, 11 (4.3%) refused to participate, 5 said they had little time available, and 6 did not provide explanations. Of the people who refused, 7 were nurses (6.7% of the total number of nurses), 1 psychiatrist (2.7% of the total number of psychiatrists), 1 psychologist (5.5% of the total category), 1 doctor in training (6.25% of the total number of doctors in training), and 1 belonging to the “other” category (5.5% of the total). The profile of MHW recruited was: psychiatrists, 37 (15.4%); medical doctors (psychiatrist in training), 16 (6.7%); nurses, 105 (43.75%); clinical psychologist and psychologists in training, 18 (7.5%); educators, occupational therapists, and technicians of rehabilitation, 24 (10%); socio-health staff involved in personal care, 20 (8.3%); and others, 20 (8.3%).

Of the 219 users contacted, 19 (8.7%) refused to participate, 8 (3.6%) said they had little time available, 2 interrupted the compilation because they declared that the questions were too complex (0.9%), and the 9 others (4.1%) did not provide explanations.

The demographic characteristics of the study sample are shown in Table 1.

The two sub-samples are strongly unbalanced by gender, with fewer males (28.3% vs. 60%, OR 0.26; CI95% 0.18–0.39); by age, with more old adults (>49, 53.3% vs. 35.5%, OR = 2.08, CI95% 1.41–3.05); and by institution, with more graduates (53.75% vs. 9.2, OR = 9.40, CI95% 5.64–15.6) among health workers.

Table 2 shows the comparison of the answers in items 1–6 of WWRR about health workers and the users of the mental health services of South Sardinia. Users of mental health services show a better satisfaction with the care received in comparison with the work satisfaction of mental health workers (Item 1, mean score 5.14 ± 1.17 of users against 4.43 ± 0.98 of mental health workers, F = 52.167, *p* < 0.0001); a better perception that the users of the service are satisfied with the care received (Item 2; 5.26 ± 0.99 against 4.43 ± 1.20, F = 61.048, *p* < 0.0001); a better satisfaction about the organizational aspects of care (or work for professionals) in the facilities (Item 3; 5.14 ± 1.10 users against 3.91 ± 1.32 professionals, F = 109.992, *p* < 0.0001); a similar perception about the respect of the human rights of the people cared for in the facilities (Item 4, mean score 5.26 ± 1.10 users against 5.09 ± 1.13 workers, F = 2.529, *p* = 0.112); and a better perception that the human rights of health professionals working in mental health are respected (Item 5, mean score 4.89 ± 1.22 against 4.51 ± 1.29, F = 19.430, *p* < 0.0001).

With regard to item 6 of the WWRR, on the perceived satisfaction with resources for care, users showed a more optimistic evaluation (mean score 2.68 ± 1.10 of users against 3.29 ± 0.85, F = 9.943, *p* = 0.002). Table 3 shows the results of Item 7 of the WWRR, where some differences between the needs of medical doctors and social workers were found between users and health workers, with users showing less need for doctors (27.1% staff vs. 13.5% users, OR staff = 2.38, CI95% 1.45–3.90) and a higher need for social workers (2.5% health workers vs. 7.5%, OR staff = 0.26 CI95% 0.09–0.73). No other differences emerged in the perception of the need for other professional figures by both users and health workers, who in a large percentage were convinced that psychologists are missing (26.4 health workers vs. 27.5% users, OR = 0.86, CI95% 0.56–1.32). The other figures, including nurses (8.8 vs. 7.5, OR = 1.18, CI 95% 0.60–2.36) and security officers (2.9 vs. 2.5, OR = 1.17, CI95% 0.37–3.75), seem less necessary for users than health workers.

## 5. Discussion

A first preliminary consideration concerns the imbalance of the samples, which seems, even based on the literature, a consequence of the characteristics of the two different populations of origin. In fact, it is known that users of mental health services, even those in Sardinia, have a low level of education (Carta et al., 2013), while health workers in Italy frequently graduate from universities (doctors and younger nurses). It is also known that among health workers in Italy there is an excess of female staff (63.8% in 2010 with an increasing trend) and that the same staff tend to age [18]. For this reason, we did not carry out the statistical analysis adjusting by demographic characteristics, even if the two groups were unbalanced, because, as explained, the differences in the demographic characteristics were typical of the population from which the samples were extracted.

The study found that the users of mental health services of an Italian region show a higher level of satisfaction with the care received than health workers of the same region with their work. Furthermore, the users of the mental health care community network of South Sardinia, in comparison with mental health workers of the same area, show a higher level of satisfaction about the organization of the network delivering care, and are more convinced that users are satisfied with the care received and that human rights are respected by health workers in the facilities. However, when in a previous article the same sample of mental health workers was compared with workers from other health sectors in the same area (Carta et al. in press), and in another survey, a sample of mental health workers of the same area was compared with mental health workers from other countries [10], the mental health workers in South Sardinia showed higher scores on the WWRR satisfaction scale in all items except for satisfaction with available resources for care. Considering these previous comparisons and the fact that the responses of mental health workers in any case show a very high level of satisfaction, the level of appreciation of the treatments and awareness of the rights of the users of the services proves to be very high.

In the present survey, the users show more optimistic scores than the mental health workers in all WWRR items, except for Item 4 (To what extent do you believe that the human rights of the people who are cared for in your service are respected? Or in the users’ version “of the services in which you are cared”), in which the scores of the two groups did not show any differences. However, in this item both groups had mean responses above 85% of the maximum possible satisfaction.

Users show greater optimism regarding the resources available for care than professionals, but they do not show a particularly high score. It is in fact remarkable that the average score at this item was below 50% of the maximum (2.68/5, in which 1 was the best possible), while in all the other items the means exceeded or approached 5 on a 6-point coded scale (in which 6 was the best possible), thus being higher than 85% of the maximum codable. It should be noted that the item regarding satisfaction with resources for care (Item 7) was the only one in which mental health professionals gave lower average scores than those of health workers from other sectors [12] and the score was not higher than the one given in the other countries [10]. An apparent contradiction was thus highlighted: although mental health workers showed a high score in job satisfaction, well-being and the perception that the rights of both workers and users were respected, there was an evident dissatisfaction with the resources available [12]. This apparent contradiction is therefore confirmed by the users’ point of view, in the sense that the satisfaction with resources was decidedly lower than the satisfaction with the other measured parameters.

Even the perception of the kind of professional lacking in mental health care services showed similarities between users and mental health workers, but with some notable differences: users believed in a greater proportion that it is not useful to increase the number of doctors, while they believe, in a greater proportion than mental health workers, that it would be useful to increase the number of social workers. However, it must be emphasized that both groups believe that the most deficient figures (besides medical doctors for staff) are educators/technicians of rehabilitation. Furthermore, psychologists are also considered deficient by both users and health workers. Ultimately, both groups underline the need for professional figures who enhance psychosocial intervention (psychologists) and rehabilitation (rehabilitation technicians/educators); however, the needs of users seem more oriented towards increasing the number of professionals working at social integration (social workers), while health professionals would like to strengthen the figures more traditionally linked to the medical model of care (medical doctors). This difference in staff adherence to a more medical model of care, as opposed to a more psycho-social vision of the ideal care of users, is partially consistent with the results of a series of studies conducted by the same research group in the community of the same region, in which emerged a better confidence in a psychosocial model of treatments and a low confidence in psychiatrists for the care of depressive disorders [19,20].

The study, although it shows results in some exciting aspects, seems to detect, between the lines, a critical moment in the history of mental health care in Italy [21]. In fact, the Italian mental health care system still seems to provide care of an excellent perceived quality (from the point of view of the users themselves), and this is partially due the radical denial of a hospital model, as it is known that in Italy assistance is focused on community care, and a 1978 law has kept psychiatric hospitals closed for over 20 years [22,23]. Nevertheless, the mental health care system in Italy is now affected by the progressive decrease in public health expenditure, which dropped dramatically from 2008 to 2011 but never returned to pre-economic crisis levels [24]; in Italy, the per capita expenditure for health monitored in 2017 was 15% below the per capita expenditure of the average of European Union, while the Italian per capita income was practically equal to the European average [24]. Moreover, mental health care expenditure accounts for only 3% of general health expenditure, while in other European countries with a similar income to Italy, mental health care expenditure is around 8–10% of general health expenditure; therefore, other countries spend more for health than Italy. These objective data perfectly reflect the dissonance found in both operators and users between the very high scores in satisfaction, organizational well-being and respect for rights and the low level of satisfaction in the resources that can be used for treatment. It seems unlikely that this excellent result can be maintained if spending cuts continue.

It should be emphasized that the research was conducted at the time of the COVID-19 pandemic, where the fear of infection, the restrictive measures that prevented the normal inter-individual exchange and the alteration of the rhythms induced by the lockdown measures have caused a condition of stress, particularly for people with psychosocial disabilities [25,26]. The emergency condition has also reverberated in all health services, increasing barriers and making access to care problematic [27].

The proximity to a person’s needs, as it is in a totally community-oriented mental care network like in Italy, was particularly useful in the general framework of the COVID-19 pandemic, during which the interviews of this study were carried out. The treatments provided in the hospital have in fact suffered from a distance from the people, the lengthening of waiting times for treatment, the fear of being infected and from the staff being put under pressure due to the proximity to “dangerous” wards and overcrowding, as demonstrated by previous studies on the non-satisfaction of health staff not working in mental health in Italy [12], and in the strong unmet needs that emerged in more hospital-centered mental health care settings during the COVID-19 pandemic [28].

The stress on health staff caused by the COVID-19 pandemic was certainly less impactful in small units scattered throughout the territory with staff composed of a few professionals well known by users, citizens and institutions. Precisely because of the closer knowledge of the users and their life context, the use of remote support such as telemedicine or social media was easier, and also for family members and caregivers, maintaining the possibility of “face-to-face” visits for emergencies [29,30,31]. The rooting of the teams in the social framework has allowed a better monitoring of critical situations, even with distance, with the help of formal (such as social units of the municipalities) and informal supports (such as family, friends and voluntary networks).

On the other hand, this study identified a condition of well-being in users and staff in mental health care in Italy during COVID-19, which is in line with surveys conducted in Italy in the same era that found a condition of well-being in mental health workers [32,33]. The consistency of the good results from the other studies in Italy in terms of mental health during COVID-19 [32,33] can also support the generalizability of the Sardinian results to the Italian model, which, albeit with differences, is nevertheless unique in its specificity.

These excellent results on users’ and MHW’ satisfaction were recorded in Italy, while the health staff of non-psychiatric agencies experienced stress and frustration [34]; Lasalvia et al. [35] found that 53.8% (95% CI [51.0–56.6%]) of health care workers showed symptoms of post-traumatic distress, 50.1% (95% CI [47.9–52.3%]) symptoms of clinically relevant anxiety and 26.6% (95% CI [24.7–28.5%]) of at least moderate depression. Similarly, Rossi et al. [36] found that 49.4% of health workers showed symptoms of post-traumatic stress disorder, 24.7% symptoms of depression, and 19.8% symptoms of anxiety; at the same time there was a substantial distrust of the population towards the health system as a whole [37]. Future studies will have to confirm the hypothesis that an organizational model centered on community care in mental health can respect citizens’ rights to a greater extent and work well even in the conditions of over-stress due to the pandemic.

## 6. Limits

The main limitations of this study are the self-selected and non-probabilistic nature of the sample without randomization, and the limitation of focusing on only one region of the country.

However, this study, due to the relevance of its results, can be a starting point for multicentered, transnational research conducted on more representative samples of specific realities.

The study was not originally intended to assess the condition under COVID-19. The pandemic makes the results more interesting, although introducing a further variable makes the subsequent hypothesis eminently heuristic and in need of confirmation. That we do not know what an exceptional condition the pandemic represents is because it is turning into an endemic; this is one more reason to recommend further studies to confirm the evidence shown.

## Figures and Tables

**Table 1 jcm-11-00997-t001:** Socio-demographic characteristics of the study samples.

		Mental Health Professional N (%)	Users N (%)	Odd Ratio
Gender	Men	68 (28.3)	120 (60)	0.26 (CI95% 0.18–0.39)
Age	>49	128 (53.3)	71 (35.5)	2.08 (CI95% 1.41–3.05)
Education	Degree	129 (53.75)	22 (11)	9.40 (CI95% 5.64–15.6)
	High school	83 (34.6)	71 (35.5)	0.96 (CI95% 0.64–1.42)
	<9 years ed.	28 (11.7)	107 (53.5)	0.11 (CI95% 0.07–0.17)
	Total	240	200	

**Table 2 jcm-11-00997-t002:** Comparison on answers at item 1-6 of WWRR about health workers and users of mental health services of South Sardinia.

	Mental Health Workers (N = 240)	Users (N = 200)	F (df 1, 438)	*p*
(1) How satisfied are you with your work? (users: of the services in which you are cared)	4.43 ± 0.98	5.14 ± 1.17	52.167	<0.0001
(2) How much you believe that the users of the service in which you work are satisfied? (users: of the services in which you are cared)	4.43 ± 1.20	5.26 ± 0.99	61.048	<0.0001
(3) How satisfied are you with the organizational aspects of your work /how your work is organized? (users: the work of the services in which you are cared)	3.91 ± 1.32	5.14 ± 1.10	109.992	<0.0001
(4) To what extent do you believe that the human rights of the people who are cared for in your service are respected? (users: of the services in which you are cared)	5.09 ± 1.13	5.26 ± 1.10	2.529	0.112
(5) To what extent do you believe that the human rights of the staff working in your service are respected? (users: of the services in which you are cared)	4.51 ± 1.29	4.89 ± 1.22	19.430	<0.0001
(6) How do you evaluate the current state of care in mental health in your service/ward, with reference to resources? (users: of the services in which you are cared)	3.29 ± 0.85	2.68 ± 1.10	9.943	0.002

**Table 3 jcm-11-00997-t003:** Needs for type of health workers in the service in which I work / I’m cared (Item 7 WWRR).

	Mental Health Workers N (%)	Users N (%)	Chi Square (with Yates Correction if Needed)—*p*	OR CI 95%
Nurses	21 (8.8)	15 (7.5)	0.227—*p* = 0.634	1.18 (0.60−2.36)
OSS—Professional for personal care	12 (5)	9 (4.5)	0.060—*p* = 0.806	1.17 (0.46−2.70)
Medical Doctors	65 (27.1)	27 (13.5)	12.171—*p* < 0.0001	2.38 (1.45−3.90)
Psychologists	59 (26.4)	55 (27.5)	0.48—*p* = 0.487	0.86 (0.56−1.32)
Occupational Therapists/Educators/ Technicians of Rehabilitation	72 (30)	70 (35)	1.25—*p* = 1.264	0.80 (0.53−1.19)
Social Workers	5 (2.5)	15 (7.5)	7.37 *—*p* = 0.013	0.26 (0.09−0.73)
Staff Security	7 (2.9)	5 (2.5)	0.001 *—*p* = 0.999	1.17 (0.37−3.75)
None needs to be incremented	0	4 (2)	2.878 *—*p* = 0.090	0 (NC)

* With Yate’s correction.

## Data Availability

All data generated or analyzed during this study are included in this published article.

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
