# Peer review of "Why Was the Perception of Human Rights Respect and Care Satisfaction So High in Users of Italian Mental Health Services during the COVID-19 Pandemic?"

_jcm, 2022, doi:10.3390/jcm11040997_

Round 1
Reviewer 1 Report
Thank you for inviting me to this manuscript. My comments are as follows.
#1 Abstract
1st line: “The compare users/mental health workers (MHW) perception…” seems typo, should change “To” instead of “The”?
7th line: Response were higher for users, but users and staff…; does the “staff” refer to “MHW”?
#2 Introduction 2nd paragraph: Please spell out WWRR in first appearance on the main text.
I understood that previous studies have reported that respect for people/users improves the well-being of the staffs (MHW) as well, however, it was difficult to follow the reason. Were there any hypothesis or discussion in those previous reports or unknown?
#3 Throughout the main text, it was quite confusing if authors are referring the same thing by using those terms such as “mental health workers/MHW”, “health workers”, “staff”, “professionals”.
#4 Methods: I could not find the detail of how the researchers recruited the participants. Were the users asked to participate on their visit to mental health centers? What was the inclusion/exclusion criteria? How was the consent for participation obtained from each participants? What kind of occupation does the medical worker specifically refer to here (From the results, it seems to refer to nurses, OSS, doctors, psychologists, OT, Social Workers, etc. is this correct?) I think it needs to elaborate who and how about the enrollment of the study.
# Methods-Statistical Analysis: The result shows the ORs When comparing the two group. I could not find were the ORs came from that I think the authors have not fully described the analysis procedure.
# Results: Consistent to the comment I have addressed for the method, how many were the potential participant and were there any who refused to participate and/or met the exclusion criteria? There could be a flow diagram of participation/response with response rate if possible.
# Results-Table1: The display of the table is difficult to understand. I think it is better to describe all items without omitting them (i.e. instead of addressing only men on gender, describe “Women” as well) and the total in the bottom line could be omitted with regard to readability.
# Table2: I think it the rows and columns should be reversed. In addition, I missed whether the authors have adjusted by demographic characteristics which was unbalanced between the two groups. I recommend to perform an analysis adjusted for background factors, or if not, please elaborate the reason.
# Discussion: Most of the results and discussion of them seems there is nothing to do with COVID pandemic. It is indeed mentioned in the end, however how the pandemic was relevant to the result of this study remains vague and gives impression that it was one of the limitation of the study that it was conducted under the extraordinary time of era, but not the main purpose of the study. Please reconsider the position of COVID in this research paper.
Reviewer 2 Report
Gratulujemy bardzo ciekawego badania pacjentów i personelu podczas pandemii COVID 19. Prawo do pełnego uczestnictwa w życiu politycznym i publicznym osób z zaburzeniami psychicznymi jest stale zagrożone stygmatyzacją i dyskryminacją. WHO naruszając prawa osób z zaburzeniami psychicznymi ogłosiła "globalny kryzys" i "nierozwiązany kryzys globalny". W dobie COVID 19 dyskryminacja w świadczeniu świadczeń zdrowotnych obejmuje zarówno dostępność świadczeń medycznych dla pacjentów, jak i samopoczucie pracowników służby zdrowia, co ma istotny wpływ na jakość opieki zdrowotnej – co Potwierdzają Państwa badania Badanie – jak stwierdzono w artykule – ze względu na dokładność wyników, może stanowić punkt wyjścia dla wieloośrodkowych, ponadnarodowych badań prowadzonych na bardziej reprezentatywnych próbach konkretnych realiów. Wzywam państwa do dalszego działania w UE. Gratuluję artykułu.
Congratulations on a very interesting study of patients and staff during the COVID 19 pandemic. The right to full political and public participation of people with mental disorders is constantly threatened by stigma and discrimination. WHO, in violation of the rights of people with mental disorders, announced "global crisis" and "unresolved global crisis". In the era of COVID 19, discrimination in the provision of health services covers both the availability of medical services for patients and the well-being of health care workers, which has a significant impact on the quality of health care - which is confirmed by your research - as stated in the article - due to the accuracy of the results, it can be a starting point for multi-center, transnational research conducted on more representative samples of specific realities. I urge you to continue to act in the EU. Congratulations on the article.
Round 2
Reviewer 1 Report
Thank you for revising the paper. The authors have respond to my comments/queries well, and do not have further comments, however please consider to edit English. I find quite language and stylistic issues in text.